# Telocytes/CD34+ Stromal Cells in the Normal, Hyperplastic, and Adenomatous Human Parathyroid Glands

**DOI:** 10.3390/ijms241512118

**Published:** 2023-07-28

**Authors:** Lucio Díaz-Flores, Ricardo Gutiérrez, Miriam González-Gómez, Maria del Pino García, Jose Luis Carrasco, Juan Francisco Madrid, Lucio Díaz-Flores

**Affiliations:** 1Department of Basic Medical Sciences, Faculty of Medicine, University of La Laguna, 38071 La Laguna, Spainjcarraju@ull.edu.es (J.L.C.);; 2Canary Biomedical Technology Institute, University of La Laguna, 38071 La Laguna, Spain; 3Department of Pathology, Eurofins Megalab–Hospiten Hospitals, 38100 La Laguna, Spain; 4Department of Cell Biology and Histology, School of Medicine, Campus of International Excellence “Campus Mare Nostrum”, IMIB-Arrixaca, University of Murcia, 30100 Murcia, Spain; jfmadrid@um.es

**Keywords:** telocytes, CD34^+^ stromal cells, parathyroid glands, hyperplasia of parathyroids, parathyroid adenoma, adipose tissue

## Abstract

Telocytes/CD34+ stromal cells (TCs/CD34+ SCs) have been studied in numerous organs and tissues, but their presence and characteristics in the parathyroid glands have not been explored. Using immunological and ultrastructural procedures, we assess the location, arrangement, and behavior of TCs/CD34+ SCs in normal human parathyroids, during their development and in their most frequent pathologic conditions. In normal parathyroids, TCs/CD34+ SCs show a small somatic body and long thin processes with a moniliform aspect, form labyrinthine systems, connect other neighboring TCs/CD34+ SCs, vessels, adipocytes, and parenchymal cells directly or by extracellular vesicles, and associate with collagen I. TCs/CD34+ SCs and collagen I are absent around vessels and adipocytes within parenchymal clusters. In developing parathyroids, TCs/CD34+ SC surround small parenchymal nests and adipocytes. In hyperplastic parathyroids, TCs/CD34+ SCs are prominent in some thickened internodular septa and surround small extraglandular parenchymal cell nests. TCs/CD34+ SCs are present in delimiting regions with compressed parathyroids and their capsule in adenomas but absent in most adenomatous tissue. In conclusion, TCs/CD34+ SCs are an important cellular component in the human parathyroid stroma, except around vessels within parenchymal nests. They show typical characteristics, including those of connecting cells, are present in developing parathyroids, and participate in the most frequent parathyroid pathology, including hyperplastic and adenomatous parathyroids.

## 1. Introduction

Telocytes (TCs), formerly considered interstitial-like Cajal cells, are stromal cells described by Popescu and Faussone-Pellegrini in 2010/11 [1,2]. Located in the connective tissue, TCs are identifiable and distinguishable from fibroblasts, which have a different ultrastructure and cytokine profile [3]. Under electron microscopy, TCs reveal a small oval/elongated/triangular cell body and long, branched, thin, moniliform processes, named telopodes, which extend over long distances and present alternating thin segments (podomeres) and dilated portions (podoms), containing mitochondria, endoplasmic reticulum, and caveolae [1,2]. TCs express CD34 (TCs/CD34+ SCs) and show immunophenotype heterogeneity [4], but there is no specific marker for them. However, positivity for CD34, PDGFRα, and vimentin and negativity for CD31 facilitate their observation by immunochemistry and immunofluorescence [5,6]. Several possible functions have been highlighted for TCs/CD34+ SCs in numerous tissues and organs. These functions include control of tissue homeostasis, intercellular communication and integration of tissue components, morphogenesis, and regeneration/repair [1,2,4,7,8,9,10,11,12,13,14,15,16,17,18,19,20,21,22,23]. Thus, TCs/CD34+ SCs control and regulate the extracellular matrix, form scaffolds, guarantee structural support, and create microenvironments (delimiting cells) [9,13,19,20,22,23]. They act in intercellular communication through homo- and heterocellular contacts, extracellular shedding vesicles, and paracrine molecules, as well as by their endocytic ability (internalization) [10,11,12,13,21,22]. TCs/CD34+ SCs connect with macrophages, lymphocytes, plasma cells, and mast cells and participate in immunomodulation and immunosurveillance [17,23]. The role in morphogenesis, regeneration/repair, and tumor stroma formation is based in stem cell modulation (presence in several stem cell niches), guidance of migration, and as a source of other cells [13,14,15,16,17]. Numerous studies have expanded these issues, and there are recent general reviews on the expression and possible functions of TCs/CD34+ SCs in several systems and organs [24,25,26,27,28] and, more specifically, in the female reproductive system [29], testes [30], gut [31], skeletal, cardiac and smooth muscle [32], heart [33], and dermis [34,35].

Parathyroid glands form part of the endocrine system, secrete the parathyroid hormone (parathormone), which helps to maintain calcium homeostasis, and participate in numerous functions of the organs. The parathyroid stroma is formed by (1) a connective capsule; (2) septa of variable thickness, which emerge from it and incompletely delimit parenchymal lobes; and (3) very thin tracts between densely packed parenchymal cells, arranged in nests, cords, islets, and pseudo-follicles. Blood and lymphatic vessels, unmyelinated nerve fibers, fat cells, stromal cells, and extracellular matrix, including collagen, are present in the parathyroid stroma. Although stromal (interstitial) cells of the parathyroid glands have been considered in some reports, mainly in vitro studies [36,37], TCs/CD34+ SCs have not been specifically explored in normal parathyroid glands, developing parathyroids, and most frequent parathyroid diseases, including hyperplastic parathyroids, currently classed as multiglandular parathyroid disease [38,39] and parathyroid adenoma. Knowledge of the characteristics, location, and arrangement of TCs/CD34+ SCs in these conditions is of morphogenic, pathophysiological, and experimental interest, especially for assessing their modifications during parathyroid changes and for in vitro studies.

Taking into consideration the above, the objective of this study is to assess TCs/CD34+ SCs in normal, developmental, and most frequent pathological conditions of the human parathyroid glands, including multiglandular parathyroid disease (parathyroid hyperplasia) and adenomas. For this purpose, we used conventional, immunohistochemical, and electron microscopy procedures.

## 2. Results

### 2.1. TCs/CD34+ SCs in Normal Human Parathyroid Glands

TCs/CD34+ SCs are observed in the fibrous capsule, septa, and thin tracts of the parathyroid gland, especially those around the more peripheral parenchymal clusters contacting the septa. These structures include different-sized vessels and variable amounts of adipose tissue. TCs/CD34+ SCs form a labyrinthine system, especially in capsular and septal regions, in which medium-sized vessels show CD34+ ECs, smooth muscle actin positive (SMA+) medial vascular smooth muscle cells, and two or more layers of TCs/CD34+ SCs in their adventitia (Figure 1A). Collagen I was present in this adventitial area with TCs/CD34+ SCs (Figure 1B). In these regions, the adventitial TCs/CD34+ SCs were in continuity with those in the interstitium and delimiting clusters of parenchymal cells (Figure 1A,B). The location of TCs/CD34+ SCs around small vessels and parenchymal cells was evident in the thinner trabeculae and tracts with fibrous or loose connective tissue (Figure 2 and Figure 3). TCs/CD34+ SCs and their telopodes were also present between the adipocytes in the septa and thin tracts of the parathyroid glands (Figure 2A).

TCs/CD34+ SCs showed a small, triangular, spindle-shaped, pyriform or oval somatic region (cell body) and two or more processes (telopodes) (Figure 2B,C, Figure 3A,B and Figure 4A,B), oriented in several directions, communicated with other neighboring TCs/CD34+ SCs, and arranged around and between collagen I (Figure 2C, Figure 3B and Figure 4A,B). Collagen I varied in amount and characteristics, with thick compact fibers or thin isolated fibrils, depending on whether the connective tissue in the septa and tracts was fibrous or loose (Figure 2C, Figure 3B and Figure 4A,B). Frequently, the telopodes of the same TC/CD34+SC appeared in proximity to parenchymal cells, vessels, and adipocytes, totally or partially connecting these tissue components. The thickness of these cells depended on the section plane. Generally, the cell body and particularly the telopodes of TCs/CD34+ SCs were very thin. Occasionally, the parts of the cells oriented according to the cut were flat, with a lamellar or velamentous aspect (Figure 4A,B).

Ultrastructurally, the cellular body of TCs presented scant cytoplasm, with few organelles and a relatively large nucleus, showing patches of heterochromatin located close to the nuclear membrane (Figure 4C). Long and moniliform, with slender and dilated portions, the telopodes were observed around vessels and parenchymal cells, predominantly chief (principal) cells with intracytoplasmic electron-dense granules (Figure 4C). Groups of oxyphilic cells were also present, and water-clear cells were scarce in the normal parathyroid gland. Extracellular vesicles (predominantly multivesicular bodies) were seen in proximity to telopodes and parenchymal cells (Figure 4C and insert).

Small vessels within the parenchymal cell clusters, predominantly in areas furthest from the connective septa, were devoid of surrounding TCs/CD34+ SCs (Figure 5A–C). TCs/CD34+ SCs were also absent around the adipocytes associated with these small vessels within the parenchymal clusters (Figure 5C). Collagen I was not observed or decreased in the areas in which TCs/CD34SCs were absent or scarce (Figure 5B,C). Transitional regions with and without perivascular TCs/CD34+ SCs and collagen I were also seen (Figure 6A).

### 2.2. CD34+ SCs/TCs during Parathyroid Gland Development

Prominent TCs/CD34+ SCs were observed in developing parathyroids and were an important cellular component of the gland stroma, together with adipocytes, ECs, and pericytes. Long telopodes of TCs/CD34+ SCs were seen around parathyroid parenchymal cell islets located in adipose tissue (Figure 6B). The telopodes connected vessels, parenchymal cells, and adipocytes (Figure 6B). TCs/CD34+ SCs with a bipolar aspect were also seen arranged around parenchymal islets (Figure 6C).

### 2.3. TCs/CD34+ SCs in Multiglandular Parathyroid Disease (Hyperplastic Parathyroid Glands)

Conventional techniques (hematoxylin eosin staining) revealed stromal (interstitial) cells in the increased and thickened connective tissue of some septa around multiple parenchymal nodules of the hyperplastic parathyroid gland. The connective tissue in the septa was fibrous, loose, or myxoid (Figure 7A–E). The septa contained different-sized vessels, but the smallest, with thin walls, predominated in loose or myxoid connective tissue (Figure 7D,E). No differences were found in the tracts according to the prevalence of parathyroid cell type in the parenchymal clusters, and, as in normal conditions, the vessels within parenchymal nests did not show perivascular TCs/CD34+ SCs. Stromal cells were TCs/CD34+ SCs, with a predominantly bipolar morphology in septa with a fibrous aspect (Figure 7F,G) and stellate in loose or myxoid septa (Figure 7H). TCs/CD34+ SCs were also observed around small nests of parenchymal cells and adipocytes in the adipose tissue surrounding the hyperplastic parathyroid glands (TCs/CD34+ SCs around extra-glandular parenchymal nests in the adipose tissue) (Figure 8).

The arrangement of TCs/CD34+ SCs around parenchymal cells and vessels observed by immunochemistry (Figure 9A) was evidenced by electron microscopy. Ultrastructurally, thin telopodes of telocytes were seen around small vessels and in the vicinity of the chief, eosinophilic, and water-clear cells (Figure 9B–D), irrespective of their modifications in quantity, different organelle development, or presence or absence of lysosomes with variable components (Figure 9 and Figure 10A). Isolated cells, with abundant secretory granules, were also observed between telopodes of telocytes and capillaries with very thin fenestrated endothelium (Figure 10B). Long and slender cellular processes were seen, with occasional secretory granules, in close proximity to the isolated cells. In addition, extracellular vesicles were observed between these components and telopodes of telocytes (Figure 11A,B). Occasional dense granules were also present in the stroma underlying parenchymal cells (Figure 11C).

### 2.4. TCs/CD34+ SCs in Parathyroid Adenomas

TCs/CD34+ SCs and their processes were observed in the well-defined delimiting region between the adenomas and the peripheral compressed parathyroid gland, and around adenomatous nests and medium- and small-sized vessels contacting this delimiting region (Figure 12A–C). Except for these regions, most adenomatous tissue was devoid of TCs/CD34+ SCs. Thus, small thin-walled vessels without perivascular TCs/CD34+ SCs and in proximity to parenchymal cells were seen within masses of compact parenchymal nests (Figure 12C). The absence of TCs/CD34+ SCs was accompanied by decreased collagen I (Figure 12D). In the peripheral compressed parathyroid gland, fibrosis of the adventitia was observed in arterial vessels, with presence of collagen I between TCs/CD34+ SCs (Figure 13A). In addition, TCs/CD34+ SCs appeared extending their processes between vessels, collagen I, and adipocytes in the increased connective and adipose tissues of the compressed parathyroid gland (Figure 13B,C).

## 3. Discussion

In this work, we contributed for the first time the location, characteristics, and arrangement of TCs/CD34+ SCs in the adult and developmental parathyroid glands and in their most frequent pathological conditions, including parathyroid hyperplasia and adenomas.

Telocytes have been identified by electron microscopy [1,2], and the ultrastructural characteristics of the stromal/interstitial cells described here, including a small somatic body and long thin processes/telopodes with a moniliform aspect, meet the requirements for consideration as such. The correlation of their ultrastructural morphology, location, and arrangement with that obtained by immunochemistry and immunofluorescence (CD34 positivity) has also facilitated their identification by these procedures (TCs/CD34+ SCs). The demonstration of flat regions in TCs/CD34+ SCs coincident with random histologic sections is due to the lamellar and velamentous morphology of these cells, highlighted in 3D reconstructions in the skin [19].

The formation of labyrinthine systems by TCs/CD34+ SCs and the arrangement of their interconnecting telopodes around vessels and parenchymal components in the parathyroid glands are concurrent findings with those already demonstrated in different organs and tissues [20,25,32,34,40,41,42,43,44] and with the role of these cells in mechanical support and intercellular relationship. In our observations, TCs/CD34+ SCs also relate the vessels and parenchymal nests with the adipocytes since adipose tissue is an important component in these glands [45,46,47]. The presence of extracellular vesicles, including multivesicular bodies near TCs/CD34+ SCs, parenchymal cells, and vessels, also concurs with that described in similar locations in other organs and tissues [21,48,49]. These findings are of functional interest, given the role of intercellular direct contacts and of extracellular vesicles providing molecular signals in intercellular communication, one of the functions hypothesized for TCs/CD34+ SCs [21,48,49,50].

The association of TCs/CD34+ SCs with collagen I in the parathyroid glands and the non-observation or decrease in collagen I in interstitial areas where TCs/CD34+ SCs are absent or scant can be related to one of the functions suggested for TCs: organization and control of the extracellular matrix, including collagen [22]. In tissues under different conditions, we have shown that type I collagen is associated with TCs/CD34+ SCs [35,51]. Verification in other organs and tissues and the study of other possible collagens associated with TCs/CD34+ SCs require further research, which may provide information for identifying these cells and specifying the role of TCs/CD34+ SCs in collagen homeostasis.

The absence of TCs/CD34+ SCs around small vessels that penetrate parenchymal cell nests in the parathyroid glands concurs with what has been described in some organic regions. Indeed, stromal cells expressing CD34 are not observed in superficial areas of the papillary dermis and of the intestinal and gallbladder mucosa [52,53]. The absence of the stromal cells around vessels within parathyroid cell nests facilitates intimate relationships between parenchymal cells and vessels and may prevent reactive stroma formation in these parathyroid regions [53]. Likewise, the formation of large cell masses in parathyroid adenomas, with compact accumulations of parenchymal cell nests containing small vessels devoid of perivascular stromal cells, may explain the absence of TCs/CD34+ SCs in large regions of them.

Some studies have demonstrated that stromal cells isolated from hyperplastic parathyroid glands have mesenchymal potential (stromal stem cells) [36,37]. TCs/CD34+ SCs can be the perivascular and interstitial cells in the connective and adipose tissues of the parathyroid with this capacity. Indeed, among the functions hypothesized for TCs/CD34+ SCs is their behavior as progenitor cells, with participation in regeneration and repair [13,14,15,24,44,52,54,55,56,57,58,59,60,61,62,63,64]. This concurs with the finding that isolated stromal stem cells from parathyroid glands of patients with secondary hyperparathyroidism have higher osteogenic differentiation ability than bone-marrow-derived stromal stem cells [36]. It is important to note that the expression of CD34 may vary in some pathologic processes in vivo or after passages in vitro (progressive loss of CD34 expression with cell cycling) [7,24,65,66,67,68,69]. In this regard, the formation of the new septa dividing pseudolobules around parenchymal nests in hyperplastic parathyroid glands leads to increased stromal components. Likewise, although adipose tissue decreases in hyperplastic parathyroid glands, one of the possible origins of the stromal cells extracted for in vitro studies could be the intraglandular adipose tissue. Indeed, the high mesenchymal potential of the adipose tissue has been widely demonstrated [70,71,72,73].

The observation of adventitial fibrosis, with abundant collagen I between TCs/CD34+ SCs, in some arterial vessels of the peripheral parathyroid glands compressed by parathyroid adenomas is an indicator of the role attributed to the adventitia as an important regulator of vascular structure and function, in normal and pathologic conditions [74,75,76,77,78,79]. Resident stromal cells in the adventitia and bone-marrow-derived collagen I+ CD45+ circulating fibrocytes have been considered to be involved in adventitial fibrosis [74,75,76,77,78,79], an important fact in hypertension [80,81]. Our results suggest the involvement of adventitial TCs/CD34+ SCs in the arterial fibrotic processes, which requires future studies.

The existence of parenchymal cell nests in the adipose tissue surrounding hyperplastic glands (extraglandular parathyroid nests) has been flagged as an important consideration during parathyroidectomy. The presence of these extraglandular nests has been explained by the hyperplastic parenchyma push out during nodular growth, together with adipose tissue [82]. We observe TCs/CD34+ SCs around these extraglandular nests, vessels, and adipocytes, which raises an unresolved question in this work: whether they originate in the same way as parenchymal cells and adipocytes, from the perivascular and interstitial cells resident in the receptor tissue, or from circulating progenitor cells.

The similarities and differences of TCs/CD34+ SCs in normal adult and developmental parathyroid glands, as well as in parathyroid hyperplasia and parathyroid adenoma, are summarized in Table 1. In the latter, we have taken into consideration the stromal features characterizing the parathyroid glands in the aforementioned conditions. Although no differences were found in relation to the prevalence of parathyroid cell type in the multiglandular parathyroid disease (former hyperplasia) and parathyroid adenomas, considering the remarkable heterogeneity of the parenchymal phenotype described in these lesions [83] and that TCs/CD34+ SCs are in contact with parenchyma, future quantitative explorations of the numbers and density of TCs/CD34+ SCs are required.

In conclusion, TCs/CD34+ SCs are an important stromal cell component in the normal parathyroid glands and participate in pathologic processes such as hyperplasia and adenomas of these glands. In this regard, the remarkable findings and considerations are the following. (a) TCs/CD34+ SCs are present in the capsule, septa, and thin connective tracts of the normal parathyroid glands, but are absent around most vessels within the parenchymal nests. The presence or absence of perivascular TCs/CD34+ SCs depending on the regions of the gland may influence the pathophysiological stromal responses in them. (b) Their telopodes form labyrinthine systems and extend around vessels, adipocytes, and parenchymal cell nests, which, together with the formation of extracellular vesicles, contribute to the hypothesis of their being “connecting cells.” (c) They are prominent in developing parathyroids and connect parenchymal cells, vessels, and adipocytes. (d) TCs/CD34+ SCs associate with collagen I, which is increased and surrounded by these cells in the adventitia of vessels in parathyroid glands compressed by parathyroid adenomas. (e) They surround extraglandular parenchymal nests in the adipose tissue around hyperplastic parathyroid glands. (f) TCs/CD34+ SCs are generally absent around vessels within the hyperplastic and adenomatous parenchyma but present in the surrounding stroma, including peripheral parenchymal nests. All these findings provide a basis for future studies of these cells in the parathyroid glands.

## 4. Materials and Methods

### 4.1. Human Tissue Samples

The archives of Histology and Anatomical Pathology of the Department of Basic Sciences of La Laguna University Hospital and Eurofins Megalab-Hospiten Hospital of the Canary Islands were searched for human parathyroid gland samples in which the stroma was suitable to be examined under different conditions of the glands. In total, 37 cases were studied, in which the parathyroids were obtained by parathyroid surgery, accidental removal during thyroid surgery, adult autopsies, and fetuses received for examination following miscarriage. There were 6 that were normal, 9 developing glands (between eighth and twentieth week of gestation), 8 multiglandular parathyroid diseases (hyperplasia), and 14 adenomas. Parathyroid carcinomas were not obtained. Immunochemistry procedures and immunofluorescence in confocal microscopy were used in all cases, and 2 normal and 2 hyperplastic parathyroids were used for electron microscopy. Ethical approval for this study was obtained from the Ethics Committee of La Laguna University (Comité de Ética de la Investigación y de Bienestar Animal, CEIBA CEIBA2023-3314), including the dissociation of the samples from any information that could identify the patients. The authors, therefore, had no access to identifiable patient information.

### 4.2. Light Microscopy

The specimens were fixed in a buffered, neutral 4% formaldehyde solution, embedded in paraffin, and cut into 3 µm thick sections. The sections were deparaffinized, hydrated, and stained with hematoxylin and eosin.

### 4.3. Immunohistochemistry and Immunofluorescence

Immunohistochemistry (automated and manual procedures) and immunofluorescence were carried out as described elsewhere [24,40]. For the automated immunohistochemistry procedure, sections were incubated with the following primary antibodies: anti-CD34 (Bond™ PA0212; Leica Biosystems, Newcastle, UK) and anti-αSMA (Bond™ PA0943; Leica Biosystems, Newcastle, UK). For the double immunostaining carried out by the automated procedure, the following combination of antibodies was used: CD34/αSMA. For the nonautomated procedure, rabbit polyclonal anti-CD34 (1/100 dilution, code nº. A13929, ABclonal, Woburn, MA, USA) and mouse monoclonal anti-αSMA (1/100 dilution, code nº. ABK1-A8914, Abyntek Biopharma, Zamudio, Spain) were used. For immunofluorescence, the sections were incubated with the following primary antibodies: rabbit polyclonal; anti-CD34 (1/100 dilution, code nº. A13929, ABclonal), and anti-collagen type I (1/100 dilution, code nº. AB749P, Millipore, Burlington, MA, USA), and for the mouse monoclonal antibodies: anti-αSMA (1/100 dilution, code nº. ABK1-A8914, Abyntek Biopharma) and anti-CD34 (ready to use, class II, clone QBEnd 10, code nº. IR632, Dako, Carpinteria, CA, USA). For the double immunofluorescence labeling, sections were incubated, combining each polyclonal antibody with the monoclonal ones. As secondary antibodies, we have used those based on the primary ones and the desired color combination, being the following: biotinylated goat antimouse IgG (1:300, Calbiochem, cat. No. 401213, Calbiochem, San Diego, CA, USA), Alexa Fluor 488 goat antirabbit IgG (H + L) (1:300, cat. No. A11001, Invitrogen, San Diego, CA, USA), biotinylated goat antirabbit IgG (H + L) (1:500, Code: 65-6140, Invitrogen), Alexa Fluor 488 goat antimouse IgG (H + L) antibody (1:500, Code: A28175, Invitrogen), and Streptavidin Cy3 conjugate (1:500, Code: SA1010, Invitrogen). Nuclei were stained with DAPI (4′–6′ Diamidino-2-phenylindole, dihydrochloride) (Invitrogen, D1306, 1:5000). Fluorescence immuno-signals were obtained using a Fluoview 1000 laser scanning confocal imaging system (Olympus Optical, Tokyo, Japan). The omission of incubation in the primary antibody was used as a negative control.

### 4.4. Electron Microscopy

Electron microscopy for the ultrastructural study was carried out, as described previously [24]. The specimens were initially fixed in 2% glutaraldehyde in 100 mM sodium cacodylate buffer, pH 7.4, for 6 h at 4 °C, and subsequently washed in the same buffer, post-fixed for 2 h in 1% buffered osmium tetroxide, dehydrated through increasing concentrations of ethanol (40–100%), and embedded in Spurr’s resin. Ultrathin sections were collected on nickel grids, double-stained with uranyl acetate and lead citrate and examined with the JEOL^®^ 100B and JEM 1011 Akishima, Tokyo, Japan, electron microscopes.

## Figures and Tables

**Figure 1 ijms-24-12118-f001:**
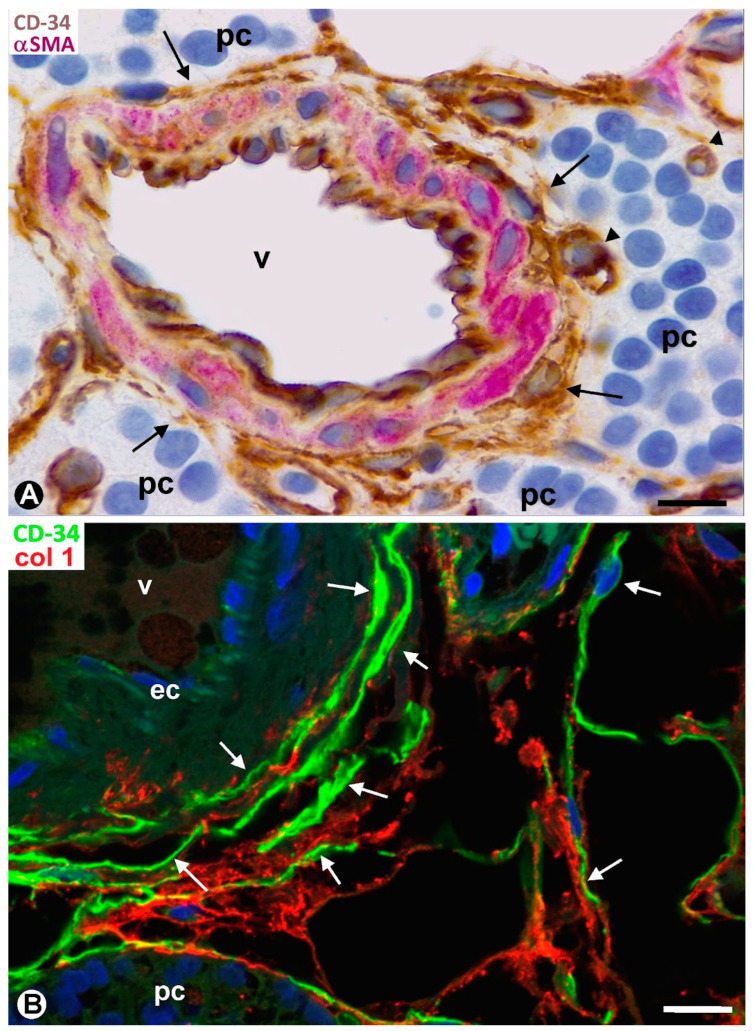
CD34+ SCs/TCs forming labyrinthine systems around medium-sized vessels (v) in the septa of a parathyroid gland. (**A**) A medium-sized vessel is observed showing CD34+ endothelial cells (brown) in the intima, αSMA+ smooth muscle cells (red) in the media, and delimiting TCs/CD34+ SCs (brown) in the adventitia. Note that TCs/CD34+ SCs adventitial cells are in continuity with TCs/CD34+ SCs in the interstitium and those around small vessels (arrowheads) and delimiting parenchymal cell clusters (arrows). (**B**) An image of a similar region observed by immunofluorescence in which endothelial cells (ec) and TCs/CD34+ SCs (arrows) express CD34 (green, ec with weak expression). Note that collagen I (red) is associated with TCs/CD34+ SCs. pc: Parenchymal cells. (**A**) Double immunochemistry for CD34 (brown) and αSMA (red). Hematoxylin counterstain. (**B**) Double immunofluorescence for CD34 (green) and collagen I (red). DAPI counterstain. Scale bars: (**A**,**B**) 15 µm.

**Figure 2 ijms-24-12118-f002:**
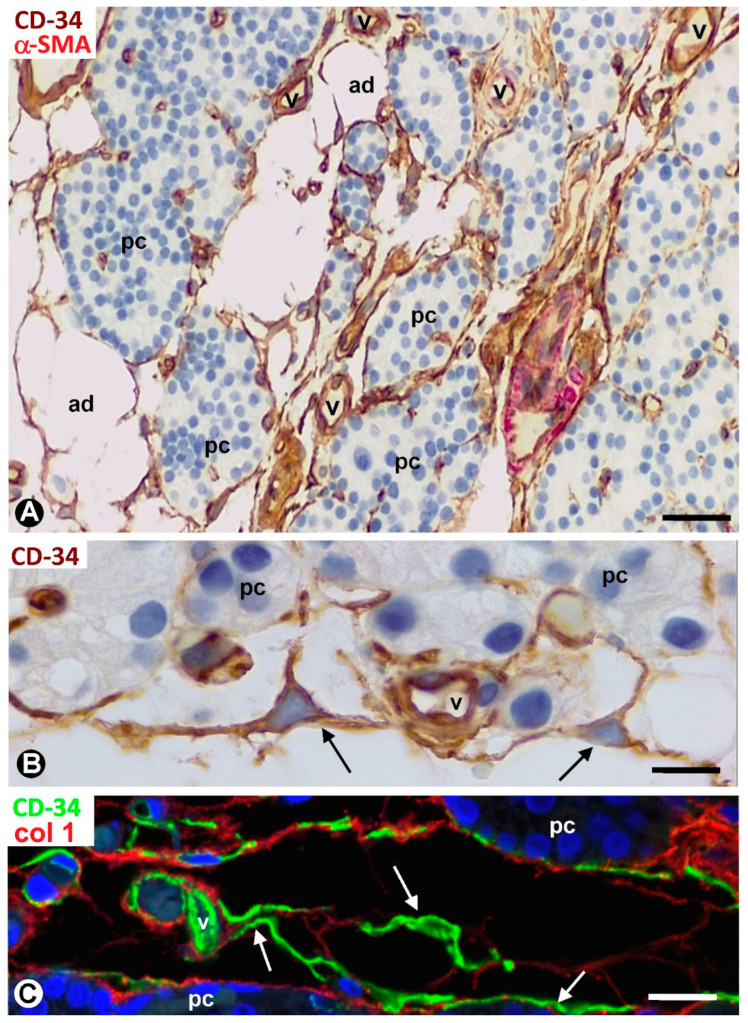
Thin connective tracts of a parathyroid gland, containing small vessels (v), TCs/CD34+ SCs (arrows, brown in (**A**,**B**) and green in (**C**)), and adipocytes (ad), arranged between parenchymal cell nests (pc). Note the stellate morphology of the TCs/CD34+ SCs in (**B**) and their relationship with collagen I in (**C**). (**A**,**B**) Double immunochemistry for CD34 (brown) and αSMA (red). Hematoxylin counterstain. (**C**) Immunofluorescence for CD34 (green) and collagen I (red). DAPI counterstain. Scale bars: (**A**) 55 µm, (**B**) 15 µm, (**C**) 20 µm.

**Figure 3 ijms-24-12118-f003:**
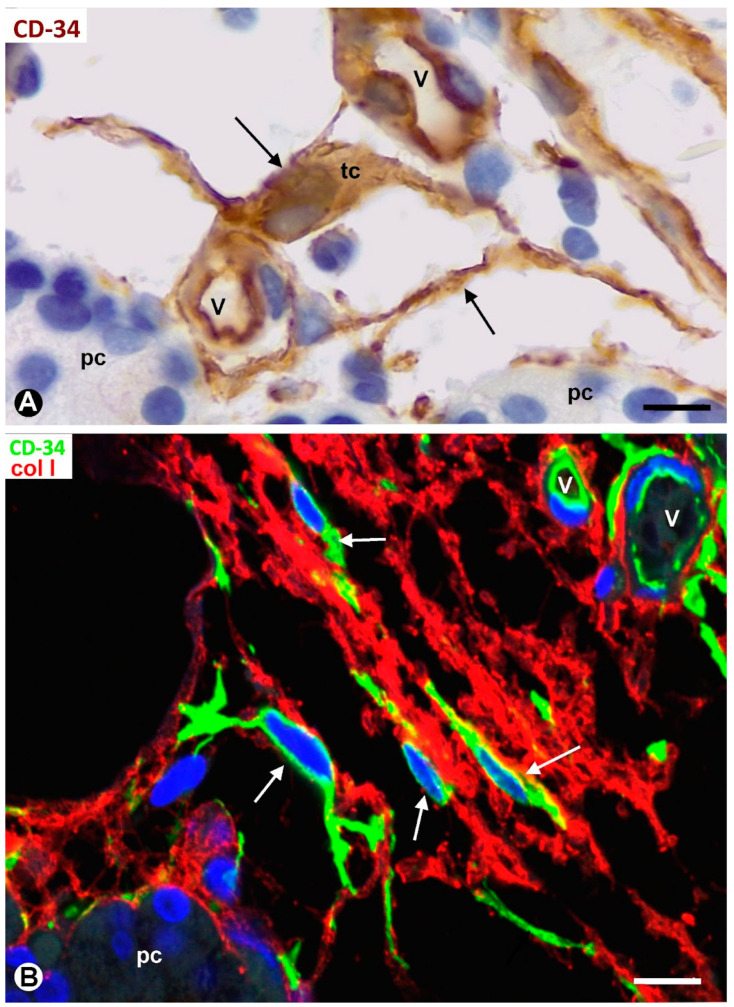
(**A**) The somatic region of a TC/CD34+SC (tc) and several telopodes (arrows) of this cell and other TCs/CD34+ SCs are observed in the interstitium and around small vessels (v) and clusters of parenchymal cells (pc) in a thin connective tract of a parathyroid gland. (**B**) TCs/CD34+ SCs and telopodes (arrows, green) are observed between a fibrous tract with thick fascicles of collagen I (red) arranged between a small vessel (v) and a cluster of parenchymal cells (pc). (**A**) Immunochemistry for CD34 (brown). Hematoxylin counterstain. (**B**) Double immunofluorescence for CD34 (green) and collagen I (red). DAPI counterstain. Scale bars: (**A**) 12 µm; (**B**) 15 µm.

**Figure 4 ijms-24-12118-f004:**
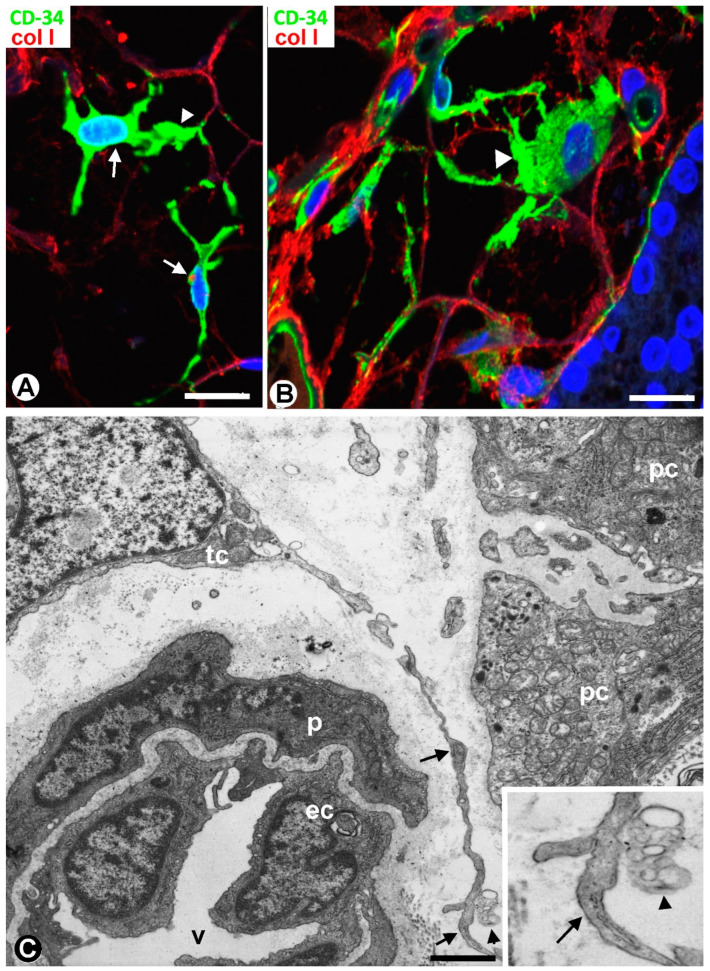
(**A**,**B**) TCs/CD34+ SCs (arrows, green) between moderate or scarce collagen I (red) in parathyroid gland thin tracts. Note that those parts of TCs/CD34+ SCs that coincide with the cut appear flat (arrowhead). (**C**) Ultrastructural demonstration of part of the somatic region of a telocyte (tc) and of a long telopode (arrow) around a vessel (v) and next to parenchymal cells (pc) of a parathyroid gland. Note the presence of a multivesicular body next to the telopode (arrowhead), presented at higher magnification in the insert. ec: Endothelial cell; p: Pericyte. (**A**,**B**) Double immunofluorescence for CD34 (green) and collagen I (red). DAPI counterstain. (**C**) Ultrathin section. Uranyl acetate and lead citrate. Scale bars: (**A**,**B**) 20 µm, (**C**) 3 µm.

**Figure 5 ijms-24-12118-f005:**
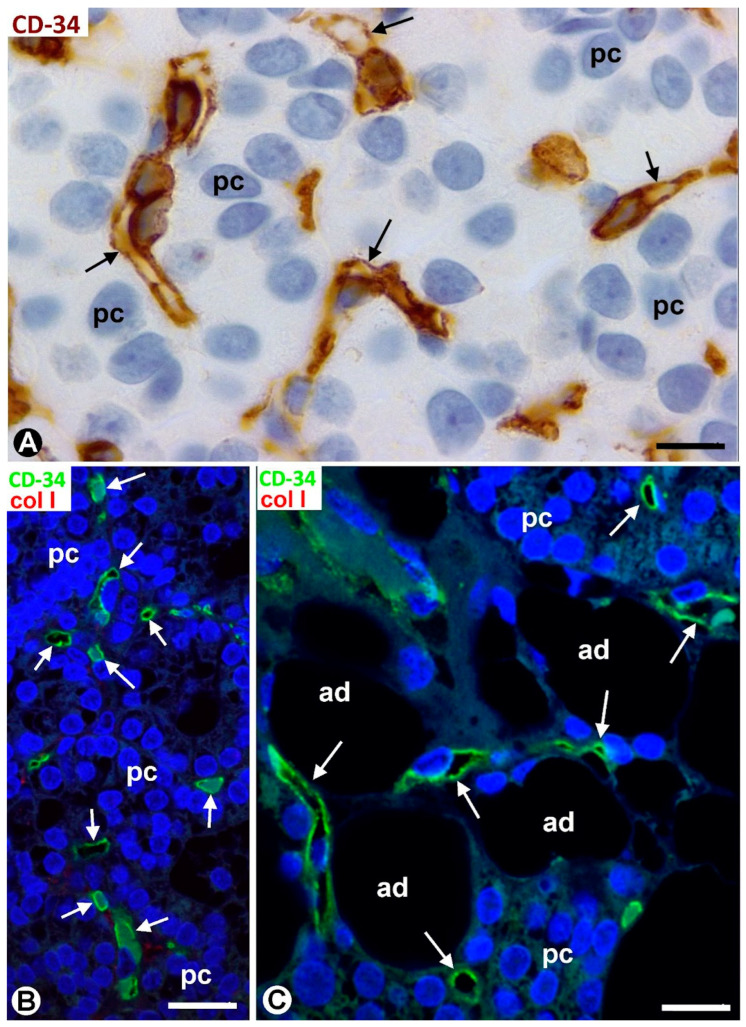
(**A**–**C**) Small vessels (arrows, brown in (**A**) and green in (**B**,**C**)), devoid of surrounding TCs/CD34+ SCs, are observed within the parenchymal cell clusters (pc). Collagen is not present in these areas in which TCs/CD34+ SCs are absent (**B**,**C**). Also note the absence of TCs/CD34+ SCs between adipocytes (ad) in these areas (**C**). (**A**) Immunochemistry for CD34 (brown). Hematoxylin counterstain. (**B**,**C**) Double immunofluorescence for CD34 (green) and collagen I (red). DAPI counterstain. Scale bars: (**A**) 12 µm, (**B**,**C**) 20 µm.

**Figure 6 ijms-24-12118-f006:**
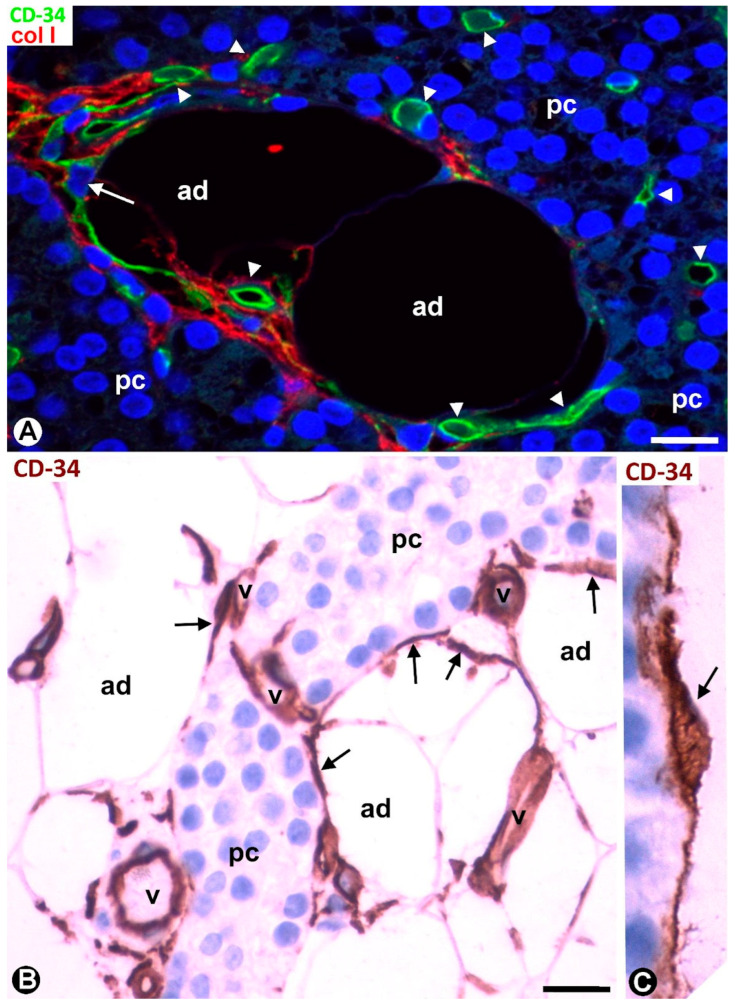
(**A**) Region of transition presenting vessels (arrowheads) with (arrow) and without perivascular TCs/CD34+ SCs and with and without interstitial collagen I (red) around parenchymal clusters (pc) and adipocytes (ad). (**B**,**C**) Developing parathyroid glands. (**B**) TCs/CD34+ SCs (arrows, brown) are observed around nests of parenchymal cells (pc), vessels (v), and adipocytes (ad). (**C**) A bipolar TC/CD34+SC (arrow) is seen around parenchymal cells. (**A**) Double immunofluorescence for CD34 (green) and collagen I (red). DAPI counterstain. (**B**,**C**) Immunochemistry for CD34 (brown). Hematoxylin counterstain. Scale bars: (**A**,**B**) 20 µm, (**C**) 1 µm.

**Figure 7 ijms-24-12118-f007:**
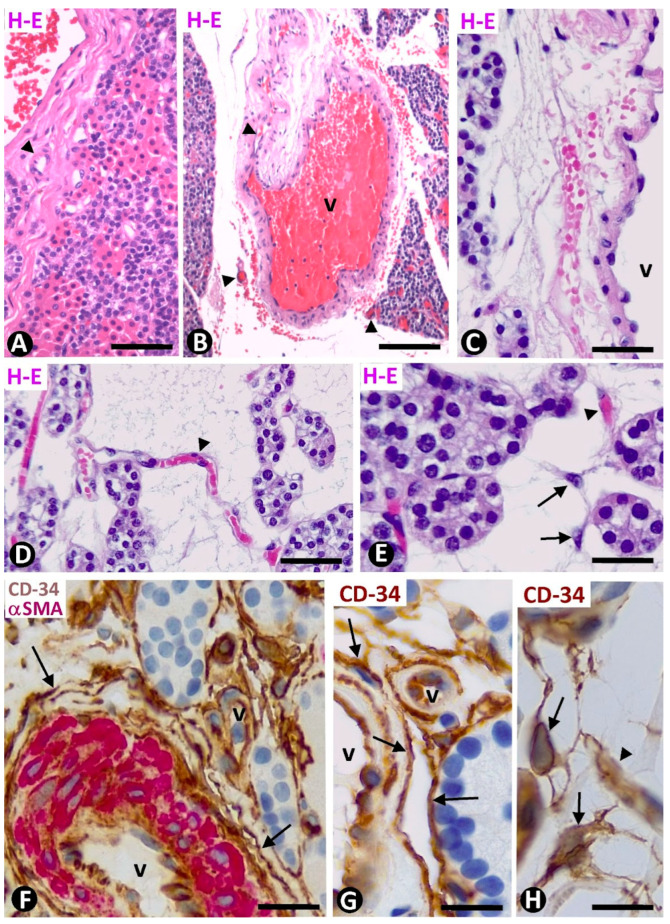
(**A**–**E**) Fibrous (**A**), loose (**B**,**C**), and myxoid (**D**,**E**) connective tissue in the thickened septa and tracts, showing large (v) and small (arrowhead) vessels in hyperplastic parathyroid glands. Note principal and oxyphilic parenchymal cells in (**A**) and stellate stromal cells in (**E**). (**F**–**H**) TCs/CD34+ SCs and their telopodes (arrows) around different-sized vessels and parenchymal cell nests. Note the predominant elongated morphology of TCs/CD34+ SCs in (**F**,**G**) and stellate in (**H**). (**A**–**E**) Hematoxylin eosin staining. (**F**): Double immunochemistry for CD34 (brown) and αSMA (red). (**G**,**H**) Immunochemistry for CD34. (**F**–**H**) Hematoxylin counterstain. Scale bars: (**A**) 80 µm, (**B**) 120 µm, (**C**,**D**) 55 µm, (**E**,**F**) 50 µm, (**G**) 20 µm, (**H**) 15 µm.

**Figure 8 ijms-24-12118-f008:**
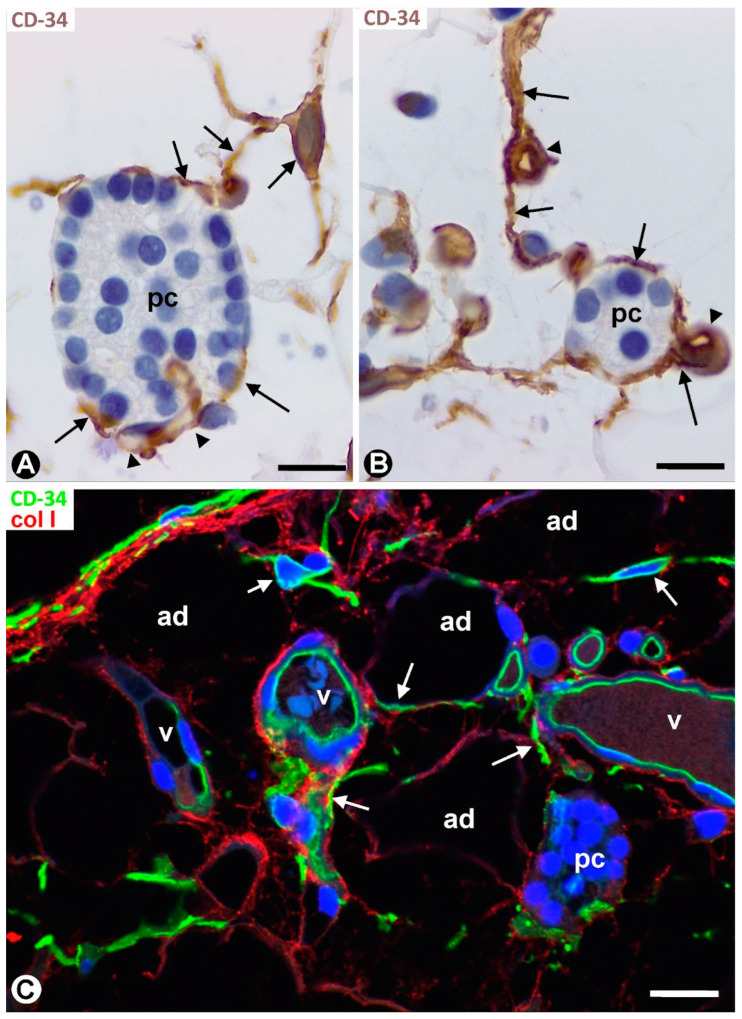
TCs/CD34+ SCs (arrows, brown in (**A**,**B**) and green in (**C**)) are present around extraglandular parenchymal cell nests (pc), vessels (v, arrowhead), and adipocytes (ad) in the adipose tissue surrounding hyperplastic parathyroid glands. (**A**,**B**) Immunochemistry for CD34 (brown). Hematoxylin counterstain. (**C**) Double immunofluorescence for CD34 (green) and collagen I (red). DAPI counterstain. Scale bars: (**A**–**C**) 20 µm.

**Figure 9 ijms-24-12118-f009:**
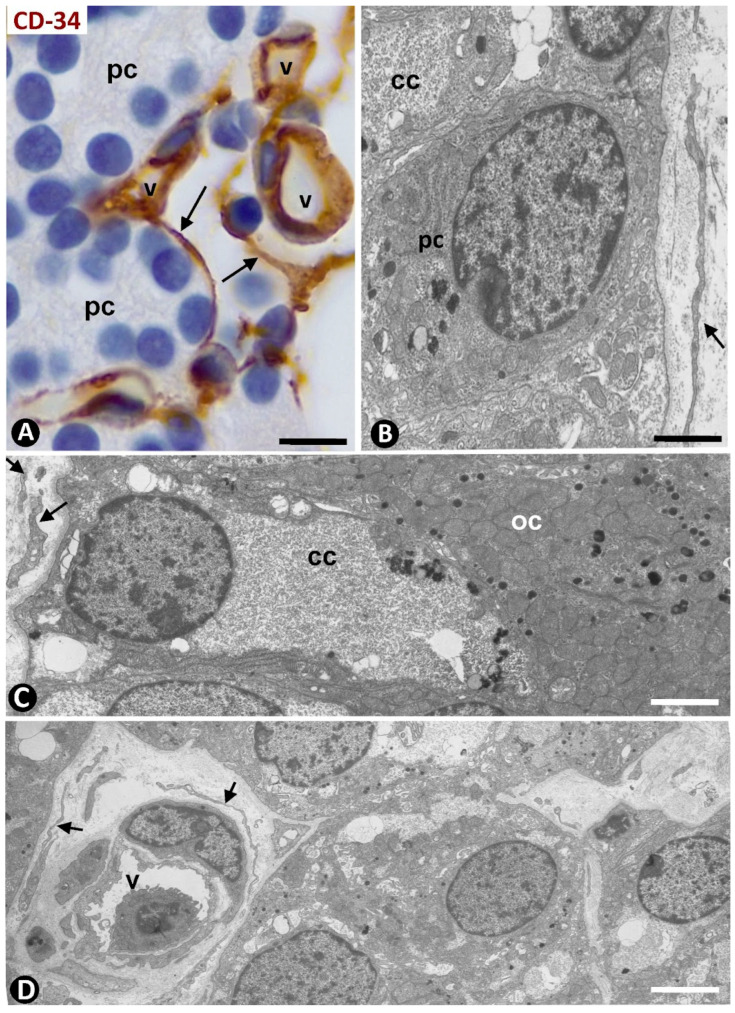
(**A**) Processes of TCs/CD34+ SCs (arrows) around parenchymal cell nests (pc) and small vessels (v) in hyperplastic parathyroids. (**B**–**D**) Ultrastructural demonstration of the arrangement of telocyte telopodes around a principal cell (pc) in (**B**), a clear cell (cc) with intracytoplasmic glycogen and an oxyphilic cell (oc) in (**C**), and a vessel (v) and several parenchymal cells in (**D**). (**A**) Immunochemistry for CD34 (brown). Hematoxylin counterstain. (**B**–**D**) Ultrathin sections. Uranyl acetate and lead citrate. Scale bars: (**A**) 15 µm, (**B**,**C**) 4 µm, (**D**) 8 µm.

**Figure 10 ijms-24-12118-f010:**
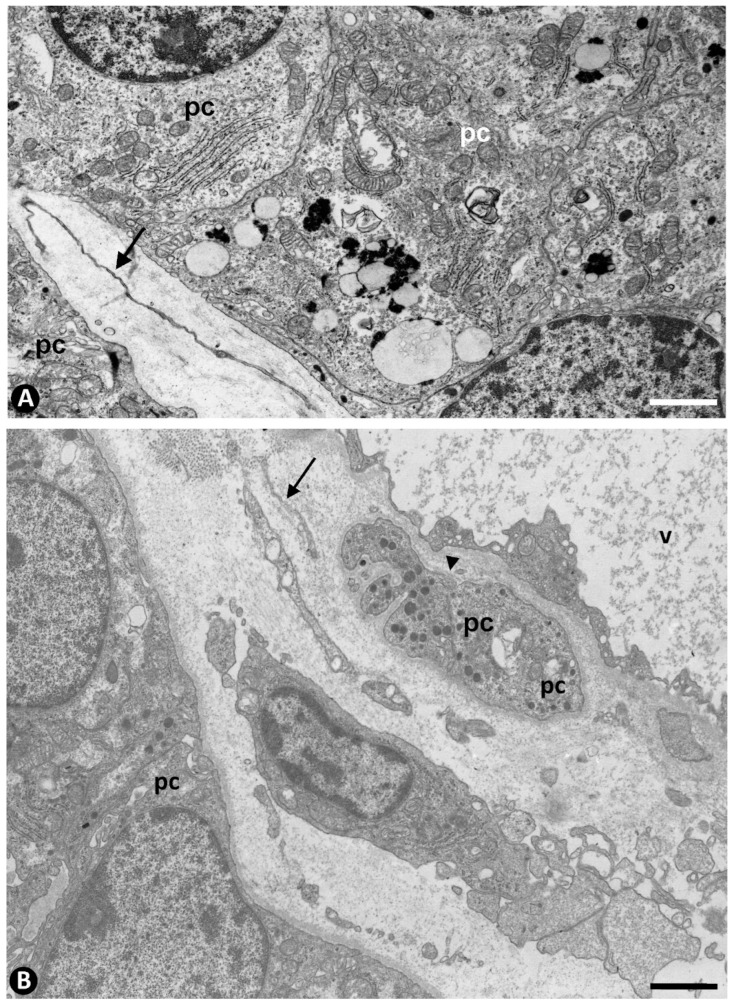
(**A**) A telocyte telopode (arrow) is observed around parenchymal cells (pc), showing scarce secretory granules, variable organelle development, and presence of lysosomes. (**B**) An isolated principal cell with secretory granules (pc, arrowhead) arranged between a fenestrated vessel (v) and telopodes (arrow). Ultrathin sections, uranyl acetate, and lead citrate. Scale bars: (**A**,**B**) 3 µm.

**Figure 11 ijms-24-12118-f011:**
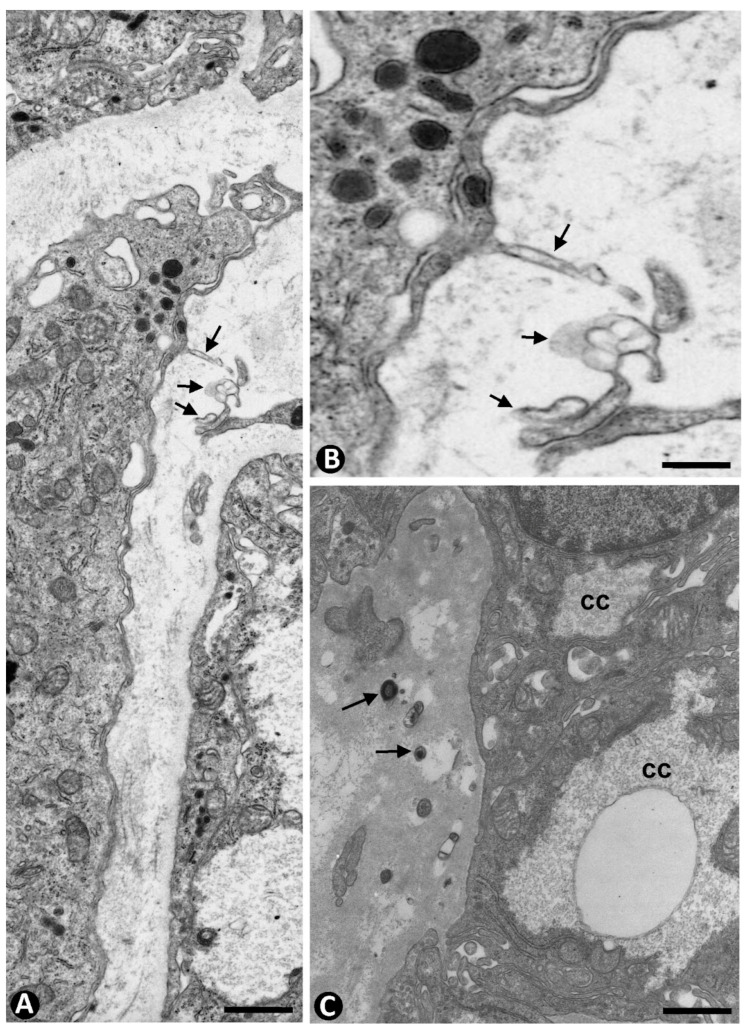
(**A**) Extracellular vesicles (arrows) are observed in the vicinity of telocytes and cells with secretory granules in hyperplastic parathyroids. (**B**) corresponds to the area of extracellular vesicles in (**A**) at higher magnification. (**C**) Dense bodies (arrows) in the stroma underlying clear cells (cc) of hyperplastic parathyroid. Ultrathin sections. Uranyl acetate and lead citrate. Scale bars: (**A**,**C**) 2 µm, (**B**) 1 µm.

**Figure 12 ijms-24-12118-f012:**
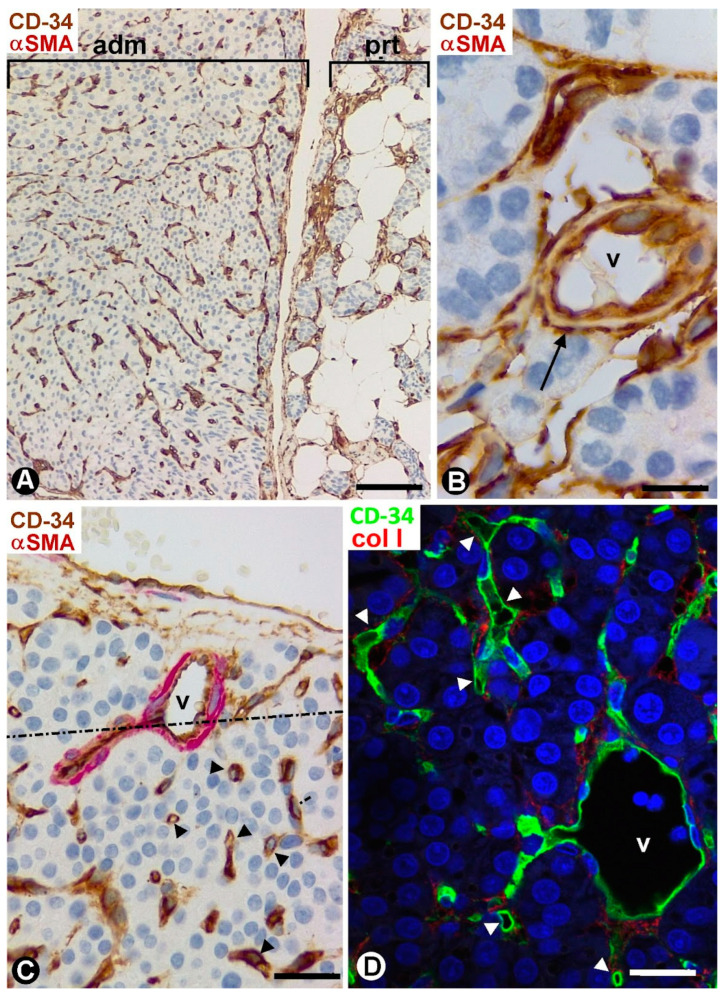
(**A**) Delimiting region between a parathyroid adenoma (adm) and the peripheral compressed parathyroid gland (prt). (**B**,**C**) TCs/CD34+ SCs (brown, arrows) are seen around vessels (v) and parenchymal nests in areas of the adenoma immediately adjacent to the delimiting region. Note the absence of TCs/CD34+ SCs around small vessels (arrowheads) away from this region (**C**). (**D**) Little or no presence of collagen I is observed where TCs/CD34+ SCs are absent. (**A**,**C**) Double immunochemistry for CD34 (brown) and αSMA (red). (**B**) Immunochemistry for CD34 (brown). (**A**–**C**) Hematoxylin counterstain. (**D**) Double immunofluorescence for CD34 (green) and collagen I (red). DAPI counterstain. Scale bars: (**A**) 120 µm, (**B**) 15 µm, (**C**) 50 µm, (**D**) 20 µm.

**Figure 13 ijms-24-12118-f013:**
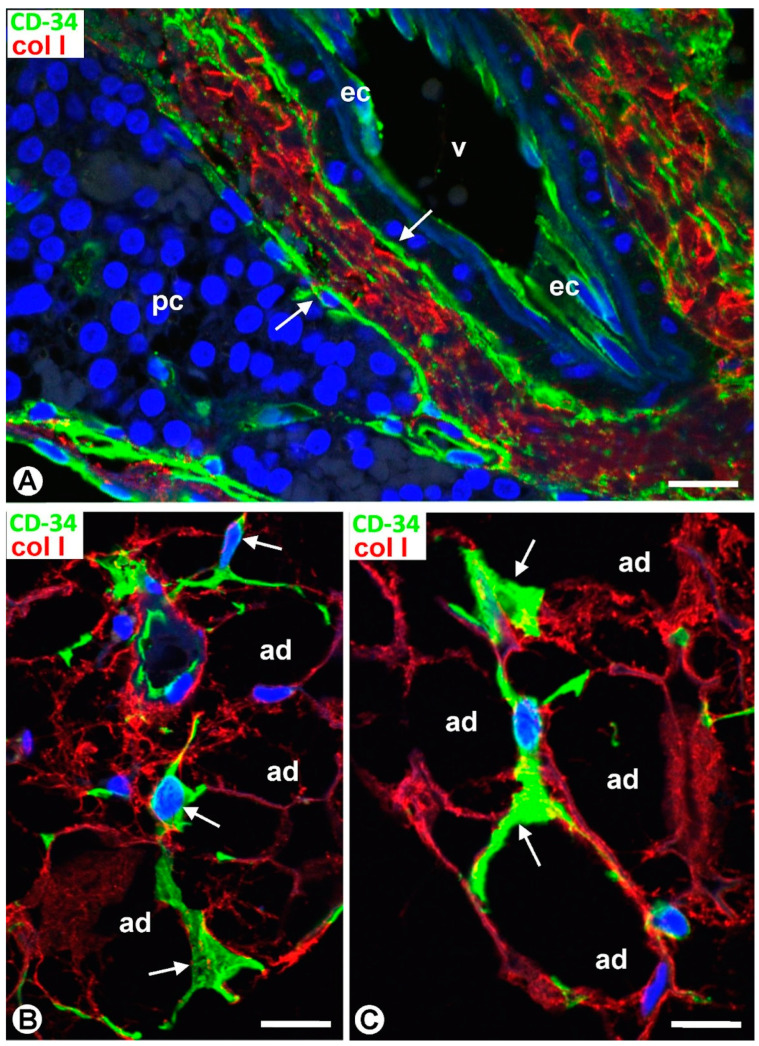
(**A**) Adventitial fibrosis with presence of collagen I (red) between TCs/CD34+ SCs (green, arrows) is observed in a vessel (v) of a peripheral parathyroid gland compressed by an adenoma. ec: endothelial cells; pc: parenchymal cells. (**B**,**C**) TCs/CD34+ SCs (green, arrows) between collagen I (red) and adipocytes (ad) in the increased adipose tissue in the compressed parathyroid gland. (**A**–**C**) Double immunofluorescence for CD34 (green) and collagen I (red). DAPI counterstain. Scale bars: (**A**–**C**) 20 µm.

**Table 1 ijms-24-12118-t001:** TCs/CD34+ SCs parathyroid glands (PG).

	Adult PG	Developing PG	Hyperplastic PG (Multiglandular Parathyroid Disease)	PG Adenoma	Compressed PG Tissue by Adenoma
Arrangement of the stroma	A—PG capsule B—Septa and thin tracts around parenchymal nests C—Adipose tissue in septa/tracts D—Vessels and adipocytes within parenchymal nests	A—Thin stromal components around small parenchymal nets and adipocytes B—Some	A—PG capsule B—Increased fibrous or loose septa around parenchymal masses and thin tracts in peripheral parenchymal nests of parenchymal masses C – Vessels within parenchymal masses	A—Thin tumor capsule B—One (occasionally two) parenchymal mass(es) with vessels within them	A—PG capsule B—Septa and tracts with increased fibrous and adipose tissues around parenchymal nests C—Vessels and adipocytes within parenchymal nests
Stromal regions with presence of TCs/CD34+ SCs	A, B, C, and around vessels and adipocytes in peripheral areas of parenchymal nests	In A.	In A, B.	In A.	A, B, and around vessels and adipocytes in peripheral areas of parenchymal nests
Stromal regions with absence of TCs/CD34+ SCs	D, absent around vessels and adipocytes within parenchymal nests, except in their periphery	In B (absent around vessels within parenchymal nests, except in their periphery)	In C (absent around vessels within parenchymal masses)	In B (absent around vessels within the parenchymal mass)	C (absent around vessels and adipocytes within parenchymal nests, except in their periphery)
Other specific findings of TCs/CD34+ SCs depending on PG conditions	-	Homocellular networks forming scaffolds around initial parenchymal nests	- TC telopodes around isolated parenchymal cells in interstitium - around extra-glandular PG tissue	Evident difference between TC/CD34+SC absence in adenoma and presence in the compressed parathyroid tissue	TCs/CD34+ SCs as an important component in adventitial fibrosis in the arteries of the compressed PG
Common features/similarities of TCs/CD34+ SCs	Small, oval, or triangular somatic body and thin moniliform telopodes Structural components Arrangement in the interstitium, around vessels and surrounding parenchymal nests. Findings of contacting cells, including homo- and hetero-cellular contacts and presence of extracellular vesicles Association with collagen I				

## Data Availability

All data are reported in the present paper.

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
