# Peer review of "Telocytes/CD34+ Stromal Cells in the Normal, Hyperplastic, and Adenomatous Human Parathyroid Glands"

_ijms, 2023, doi:10.3390/ijms241512118_

Round 1
Reviewer 1 Report
The objective of this study is to assess TCs/CD34+SCs in normal, developmental, and parathyroid hyperplasia and adenomas. For this purpose, conventional, immunohistochemical, and electron microscopy procedures were used. The paper is well written and images are good.
Major comments
Since you report not only differences in TCs/CD34+SCs but also collagen I presence, I am an important suggestion that should help the reader to easily appreciate which are the stromal features characterizing the developing parathyroid, what the adult one and what the pathological ones. Briefly, add a table resuming ALL you have seen either it is similar or different in the developing, adult and pathological parathyroids.
Minor comments
1. Pag.2, row 58. ... which helps maintain calcium..., correct: which helps to maintain calcium
2. Pag. 18, Row 409. ….. post-fixed for 2 h in 1% osmium tetroxide… specify if it was buffered
Author Response
Major comments
The following paragraph and Table 1 have been added to the Discussion:
“The similarities and differences of TCs/CD34+SCs in normal adult and developmental parathyroid glands, as well as in parathyroid hyperplasia and parathyroid adenoma, are summarized in Table 1. In the latter, we have taken into consideration the stromal features characterizing the parathyroid glands in the aforementioned conditions.”
Minor comments
“...which helps maintain calcium...” has been replaced by “...which helps to maintain calcium…”
“...post-fixed for 2h in 1% osmium tetroxide…” has been replaced by “…post-fixed for 2h in 1% buffered osmium tetroxide…”
Thank you for your kind consideration and for your help in improving this work.
Reviewer 2 Report
This paper is a detailed study of the characteristics and placement of TCs in parathyroid tissue. In addition to normal parathyroid glands, this study also examines TC in the parathyroid glands in developmental stages, thyroid hyperplasia, and adenomas. This is a very precise study using immunostaining and electron microscopy and could be useful in future studies of the parathyroid gland. Although back ground of TC was well described in introduction part, it would be helpful if the introduction included an explanation of TC expression and possible functions in other organs. 
Additionally, I feel just a little bit uncomfortable with "material" coming after "discussion".
Overall, I figured out this is a good study that provides new insights into the detailed examination of TC in various parathyroid glands.
Author Response
The first paragraph of the Introduction has been modified to provide a more precise definition of telocytes, to contribute some historical details, and to expand the expression and possible functions of TCs/CD34+SCs in different organs (the paragraph also answers reviewer 3).
“Telocytes (TCs), formerly considered interstitial-like Cajal cells, are stromal cells described by Popescu and Faussone-Pellegrini in 2010/11 [1,2]. Located in the connective tissue, TCs are identifiable and distinguishable from fibroblasts, which have a different ultrastructure and cytokine profile [3]. Under electron microscopy, TCs reveal a small oval/elongated/triangular cell body and long, branched, thin, moniliform processes, named telopodes, which extend over long distances and present alternating thin segments (podomeres) and dilated portions (podoms), containing mitochondria, endoplasmic reticulum, and caveolae [1,2]. TCs express CD34 (TCs/CD34+SCs) and show immunophenotype heterogeneity [4], but there is no specific marker for them. However, positivity for CD34, PDGFRα, and vimentin, and negativity for CD31 facilitate their observation by immunochemistry and immunofluorescence [5,6]. Several possible functions have been highlighted for TCs/CD34+SCs in numerous tissues and organs. These functions include control of tissue homeostasis, intercellular communication and integration of tissue components, morphogenesis, and regeneration/repair [1,2,4, 7-23]. Thus, TCs/CD34+SCs control and regulate the extracellular matrix, form scaffolds, guarantee structural support and create microenvironments (delimiting cells) [9,13,19,20,22,23]. They act in intercellular communication through homo- and hetero-cellular contacts, extracellular shedding vesicles, and paracrine molecules, as well as by their endocytic ability (internalization) [10-13,21,22]. TCs/CD34+SCs connect with macrophages, lymphocytes, plasma cells, and mast cells, and participate in immunomodulation and immunosurveillance [17,23]. The role in morphogenesis, regeneration/repair, and tumor stroma formation is based in stem cell modulation (presence in several stem cell niches), guidance of migration, and as a source of other cells [13-17]. Numerous studies have expanded these issues, and there are recent general reviews on the expression and possible functions of TCs/CD34+SCs in several systems and organs [24-28] and, more specifically, in the female reproductive system [29], testes [30]., gut [31]., skeletal, cardiac and smooth muscle [32], heart [33], and dermis [34,35].”
"Material and Methods" has been placed after the Introduction.
Thank you for your kind consideration and for your help in improving this work.
Reviewer 3 Report
The article “Telocytes/CD34+ Stromal Cells in the Normal, Hyperplastic, and Adenomatous Human Parathyroid Glands” represents an outstanding original morphological study of CD34-expressing cell population in normal, developing and benign hyperfunctioning parathyroid glands. Authors have successfully combined a spectrum of morphological study methods, including routine morphology by haematoxylin-eosin stain, double immunohistochemistry, double immunofluorescence and electron microscopy. This approach has resulted in a detailed description and excellent high-quality images.
There are only few remarks for potential improvements:
1) Many readers might benefit from a wider explanation of telocyte concept in the Introduction. Please, provide an exact definition of telocytes and explain the physiological and pathogenic functions of these cells in more detail (currently, only a list of these functions is provided). If authors would find this suggestion acceptable, few historical details on the development of telocyte concept might also be of major interest.
2) Please, note, that in the current WHO classification (2022), the term “parathyroid hyperplasia” has been replaced by “multiglandular parathyroid disease”. For details and reference, please, see Uljanovs et al., 2022; PMID: 35805976; DOI: 10.3390/ijms23136981.
3) In multiglandular parathyroid disease (former hyperplasia) and parathyroid adenomas, remarkable heterogeneity has been reported in regard to parenchymal immunophenotype. This manifests in clustering of cells that express certain markers, e.g., Ki-67 or p21 protein. For details and reference, please, see “Molecular profile of parathyroid tissues and tumours: a heterogeneous landscape” (2021); PMID: 34706517. Was there any heterogeneity of telocyte density observed in hyperfunctioning benign parathyroid glands, considering that telocytes are in contact with parenchyma?
4) Check the formatting of references, please. It should be in accordance with the “Instructions for Authors”.
Finally, I would like to thank the authors for their contribution. It was a pleasure and a true honour to review this manuscript.
The level of English language is generally good: the manuscript is comprehensible and truly interesting. Nevertheless, check the text for minor misprints, please. For instance, it seems that part of text is missing in the name of first affiliation. I would also advice to reconsider the terms “organic functions”, “organic regions”, “mesenchymal potential”.
Author Response
1. The first paragraph of the Introduction has been modified to provide a more precise definition of telocytes, to contribute some historical details, and to expand the expression and possible functions of TCs/CD34+SCs in different organs (the paragraph also answers reviewer 2)
“Telocytes (TCs), formerly considered interstitial-like Cajal cells, are stromal cells described by Popescu and Faussone-Pellegrini in 2010/11 [1,2]. Located in the connective tissue, TCs are identifiable and distinguishable from fibroblasts, which have a different ultrastructure and cytokine profile [3]. Under electron microscopy, TCs reveal a small oval/elongated/triangular cell body and long, branched, thin, moniliform processes, named telopodes, which extend over long distances and present alternating thin segments (podomeres) and dilated portions (podoms), containing mitochondria, endoplasmic reticulum, and caveolae [1,2]. TCs express CD34 (TCs/CD34+SCs) and show immunophenotype heterogeneity [4], but there is no specific marker for them. However, positivity for CD34, PDGFRα, and vimentin, and negativity for CD31 facilitate their observation by immunochemistry and immunofluorescence [5,6]. Several possible functions have been highlighted for TCs/CD34+SCs in numerous tissues and organs. These functions include control of tissue homeostasis, intercellular communication and integration of tissue components, morphogenesis, and regeneration/repair [1,2,4, 7-23]. Thus, TCs/CD34+SCs control and regulate the extracellular matrix, form scaffolds, guarantee structural support and create microenvironments (delimiting cells) [9,13,19,20,22,23]. They act in intercellular communication through homo- and hetero-cellular contacts, extracellular shedding vesicles, and paracrine molecules, as well as by their endocytic ability (internalization) [10-13,21,22]. TCs/CD34+SCs connect with macrophages, lymphocytes, plasma cells, and mast cells, and participate in immunomodulation and immunosurveillance [17,23]. The role in morphogenesis, regeneration/repair, and tumor stroma formation is based in stem cell modulation (presence in several stem cell niches), guidance of migration, and as a source of other cells [13-17]. Numerous studies have expanded these issues, and there are recent general reviews on the expression and possible functions of TCs/CD34+SCs in several systems and organs [24-28] and, more specifically, in the female reproductive system [29], testes [30]., gut [31]., skeletal, cardiac and smooth muscle [32], heart [33], and dermis [34,35].”
2. In the paragraph following the term “parathyroid hyperplasia” and in the sections in which it is named, we have added the more appropriate term: “multiglandular parathyroid disease”, according to the current WHO classification (2022).
“TCs/CD34+SCs have not been specifically explored in normal parathyroid glands, developing parathyroids, or most frequent parathyroid diseases, including hyperplastic parathyroids, currently classed as multiglandular parathyroid disease [38,39] and parathyroid adenoma.”
3. The similarities and differences of TCs/CD34+SCs in normal adult and developmental parathyroid glands, as well as in parathyroid hyperplasia and parathyroid adenoma, are now summarized in Table 1, in reply to reviewer 1. We have added the following to this paragraph:
“Although no differences were found in relation to the prevalence of parathyroid cell type in the multiglandular parathyroid disease (former hyperplasia) and parathyroid adenomas, considering the remarkable heterogeneity of the parenchymal phenotype described in these lesions [83] and that TCs/CD34+SCs are in contact with parenchyma, future quantitative explorations of the numbers and density of TCs/CD34+SCs are required.”
4. The format of the references and some terms of the English language have been checked according to the “Instructions for Authors”.
“Organic functions” has been replaced by “functions of several organs”.
“Organic regions” has been replaced by “in some regions of the organs”.
Thank you for your kind consideration and for your help in improving this work.
Round 2
Reviewer 1 Report
What I have asked has been done